# Relative Belief Inferences from Decision Theory

**DOI:** 10.3390/e26090786

**Published:** 2024-09-14

**Authors:** Michael Evans, Gun Ho Jang

**Affiliations:** 1Department of Statistical Sciences, University of Toronto, Toronto, ON M5G 1Z5, Canada; 2Ontario Institute for Cancer Research, MaRS Centre, 661 University Avenue, Suite 510, Toronto, ON M5G 0A3, Canada; gjang@oicr.on.ca

**Keywords:** Bayesian inference, evidential inference, statistical evidence, relative belief, loss functions, Bayesian-unbiasedness, Bayes rules, admissibility, limits of Bayes rules

## Abstract

Relative belief inferences are shown to arise as Bayes rules or limiting Bayes rules. These inferences are invariant under reparameterizations and possess a number of optimal properties. In particular, relative belief inferences are based on a direct measure of statistical evidence.

## 1. Introduction

We consider a sampling model for data *x*, as given by a collection of densities {f(·|θ):θ∈Θ} with respect to a support measure μ on a sample space X, and a proper prior, given by density π with respect to support measure ν on Θ. When the data x∈X are observed, these ingredients lead to the posterior distribution on Θ with density given by π(θ|x)=π(θ)f(x|θ)/m(x) with respect to support measure ν, where
$$m\left(x\right)=\int_{\Theta }\pi \left(\theta \right)f\left(x\;|\;\theta \right)\;\nu \left(d\theta \right)$$
is the prior predictive density of the data. In addition, there is a quantity of interest ψ=Ψ(θ), where
Ψ:Θ→Ψ(Θ)={ψ:ψ=Ψ(θ)forsomeθ∈Θ},
for which inferences, such as an estimate ψ(x) or a hypothesis assessment $H_{0}:\Psi \left(\theta \right)=\psi_{0},$ are required. Let $\pi_{\Psi }$ denote the marginal prior density of ψ and
$$m\left(x\;|\;\psi \right)=\int_{\Theta }f\left(x\;|\;\theta \right)\;\Pi \left(d\theta \;|\;\psi \right)$$
be the conditional prior predictive of the data after integrating out the nuisance parameters via the prior conditional distribution of θ given Ψ(θ)=ψ. Bayesian inferences for ψ are then based on the ingredients $\left(\left\{m\left(\cdot \;|\;\psi \right):\psi \in \Psi \left(\Theta \right)\right\},\pi_{\Psi },x\right)$ alone or by adding a loss function L. Note that the probability measure associated with *m* will be denoted by *M* and the probability measure associated with m(·|ψ) will be denoted by M(·|ψ) when these are used in the paper.

A natural question arises, namely, how are we to determine the inferences for ψ, namely, an estimate ψ(x), or assess the hypothesis $H_{0}:\Psi \left(\theta \right)=\psi_{0}$ based upon these ingredients? Several approaches have been put forward to answering this question. Two broad categories can be described, namely, the evidential/inferential approach and the behavioristic/decision-theoretic approach.

The evidential approach can be characterized as having the goal of letting the evidence in the data *x* determine the inferences and can be subdivided into frequentist, pure likelihood, and Bayesian theories. Central to this is the need to somehow characterize the concept of statistical evidence. The frequentist theory only uses the ingredients ({f(·|θ):θ∈Θ},x) together with the idea that inferences are to be graded based on their behavior in hypothetical repeated sampling experiments. Despite the impressive accomplishments of Alan Birnbaum in attempting to formulate a definition of statistical evidence, (see [1]), it is fair to say that there is still no such generally acceptable definition within the frequentist context. The pure likelihood theory is also based on the ingredients ({f(·|θ):θ∈Θ},x) but the idea of using repeated sampling characteristics to determine the inferences is dropped and the likelihood function L(·|x):Θ→[0,∞), as defined by L(θ|x)=cf(x|θ) for any positive constant c, is taken to be the proper characterization of statistical evidence. All inferences are then determined by the likelihood; for example, see the discussion in [2]. Again, there are gaps in this treatment, as it is unclear when the likelihood function provides evidence in favor of or against a particular value of θ being the true value, and it is unclear how the likelihood is to be used for marginal parameters ψ=Ψ(θ). The Bayesian approach based on the ingredients $\left(\left\{m\left(\cdot \;|\;\psi \right):\psi \in \Psi \left(\Theta \right)\right\},\pi_{\Psi },x\right)$ is more successful at characterizing statistical evidence concerning ψ through the principle of evidence, which, loosely speaking, suggests that if the data lead to the posterior probability of an event being greater than (less than) the corresponding prior probability, then there is evidence in favor of (against) the event being true. A precise statement of the principle of evidence is provided in Section 2.3. A full theory of inference based on this idea, and called relative belief, has been developed over a number of years (see [3]), and is sketched in Section 2.3. A much fuller discussion of the issues and developments within the context of the evidential approach to developing statistical theory can be found in [4].

The decision-theoretic approach can also be divided into frequentist and Bayesian theories. The frequentist approach is based on the ingredients ({f(·|θ):θ∈Θ},x) together with a loss function L:Θ×Ψ(Θ)→[0,∞) where L(θ,Ψ(θ))=0 for all θ and, generally, L(θ,ψ) represents the loss or penalty incurred when ψ is chosen as the true value of Ψ(θ). The idea then is to look for a decision procedure δ(x) that performs well with respect to the average loss or risk $R\left(\theta ,\delta \right)=\int_{X}L\left(\theta ,\delta \left(x\right)\right)f_{\theta }\left(x\right)\;\mu \left(dx\right),$ namely, choose a δ that makes R(θ,δ) small uniformly in θ. The frequentist decision theory, however, is not always successful in determining a suitable δ. The Bayesian theory of decision considers the prior risk $r\left(\delta \right)=\int_{\Theta }R\left(\theta ,\delta \right)\pi \left(\theta \right)\;\nu \left(d\theta \right)$ and is generally successful in determining a δ that minimizes r(δ); this is referred to as the Bayes rule. The Bayesian theory of decision has been axiomatized (see [5]); this provides considerable support for this approach.

If the various approaches to determining inferences all lead to more or less the same answers, then there would be little controversy, but unfortunately, this is not the case. The goal of this paper is to show that relative belief inferences can also be developed within the context of decision theory even though their primary motivation is through the characterization of statistical evidence. It is of historical relevance and interest that two of the founders of the statistical discipline, Fisher and Neyman, disagreed profoundly on the purpose of statistical analyses. Fisher saw the purpose of statistics as summarizing what the evidence in the observed data says about questions of scientific interest, while Neyman described the purpose in behavioristic or decision-theoretic terms, where the goal is to minimize average losses in repeated performances. Ref. [6] described this debate, which continues to be part of the statistical profession. The significance of the results of this paper is that it demonstrates that relative belief inference allows for a possible resolution of this conflict and, as we will now discuss, also resolves a general criticism of decision theory.

A natural requirement for statistical analysis is that all the ingredients chosen by the statistician need to be checkable against the observed data to ensure they align with the objective data (or, at least, are correctly collected). The choice of the model, prior, and loss functions are typically subjective decisions made by the analyst, and many consider such subjectivity to be at odds with the demands of science. However, both the model and the prior can be checked against the observed data to determine whether the choices made are contradicted by the data. Model checking has long been an acceptable, even necessary, part of statistical practice. In recent years, methods have been developed to check for a conflict between the prior and data. These methods determine if the prior placed the bulk of its mass in a region of the parameter space unsupported by the data as containing the true value of the parameter. Ref. [7] contained a discussion about checking for prior-data conflicts and also on what to do when a prior fails its checks. While such checking does not establish the objectivity of these elements, it at least allows the objective data to comment on the relevance of the choices made. However, it is unclear how one can check the loss function *L* using the data, and this ambiguity may be considered a flaw in decision theory, particularly for scientific applications.

There are, however loss functions that are considered intrinsic and that avoid this criticism. For example, Ref. [8] proposed using the intrinsic loss function based on a measure of distance between sampling distributions. Ref. [9] proposed using the intrinsic loss function based on the Kullback–Leibler divergence $KL(f\left(\cdot \;|\;\theta \right)\|f\left(\cdot \;|\;\theta^{\prime }\right))$ between f(·|θ) and $f(\cdot \;|\;\theta^{\prime }).$ When ψ=θ, the intrinsic loss function is given by
$$L\left(\theta ,\theta^{\prime }\right)=\min\left(KL\left(f\left(\cdot \;|\;\theta \right)\|f\left(\cdot \;|\;\theta^{\prime }\right)\right),KL\left(f\left(\cdot \;|\;\theta^{\prime }\right)\|f\left(\cdot \;|\;\theta \right)\right)\right).$$For a marginal parameter ψ, the intrinsic loss function is $L\left(\theta ,\psi \right)=\inf_{\theta^{\prime }\in \Psi^{-1}\left\{\psi \right\}}L\left(\theta ,\theta^{\prime }\right).$ These loss functions are intrinsic because they are based on the sampling model, allowing their suitability to be verified through model checking. The loss functions used to derive relative belief inferences for ψ=Ψ(θ) are based upon the prior $\pi_{\Psi },$ see Section 3 for the definitions, and so are also intrinsic and checkable against the data while checking for prior-data conflict.

In some contexts, relative belief inferences are Bayes rules, but in a general context, they are seen to arise as the limits of Bayes rules. This approach has some historical antecedents. For example, in [10], it is shown that the MLE is asymptotically a Bayes rule, but this conclusion is drawn under a fixed loss function, with increasing amounts of data and a sequence of priors. In the context discussed here, however, the data amount is fixed, as are the model and prior, but there is a sequence of loss functions, all based on a single fixed prior. The loss functions relevant for deriving relative belief inferences are similar to those used to justify maximum a posteriori (MAP) inferences. It can be demonstrated that under certain conditions, MAP inferences emerge as the limits of Bayes rules through a sequence of loss functions,
(1)$$L_{\lambda }\left(\theta ,\psi \right)=I_{B_{\lambda }^{c}\left(\psi \right)}\left(\Psi \left(\theta \right)\right)$$
where $\lambda >0,B_{\lambda }\left(\Psi \left(\theta \right)\right)$ is the ball of radius λ centered at ψ, $I_{A}$ denotes the indicator function for set *A* and, in the continuous case, the support measure on Ψ(Θ) is volume measure (see [11]). MAP inferences are not invariant under reparameterizations and such invariance can be considered as a desirable property of any inference method. Relative belief inferences, however, are invariant under reparameterizations.

Section 2 is concerned with describing the general characteristics of three approaches to deriving Bayesian inferences. Section 3 and Section 4 show how relative belief estimation and prediction inferences can be seen to arise from decision theory and Section 5 does this for credible regions and hypothesis assessment. In particular, it is shown here that relative belief estimators, as used in practice, are admissible. The contents of Section 3, Section 4 and Section 5 are original contributions by the authors that were derived some years ago but not published. Some of this discussion has appeared in Ref. [3] and is included here to provide a complete exposition of the relationship between relative belief and decision theory. All proofs of theorems and corollaries are in Appendix A, except for cases where Ψ(Θ) is finite, as these are quite straightforward and provide motivation for the more complicated contexts.

It should be emphasized that the authors do not consider the fact that relative belief inferences can be derived within the context of decision theory as the primary justification for the approach. Rather, the justification lies within the Bayesian context, which leads—via the principle of evidence—to a clear characterization of statistical evidence. The specific loss functions used, while appealing, are not essential to this characterization. The fact that relative belief inferences are consistent with two of the major themes of statistical research pursued over the years, in our view, provides substantial support for their appropriateness.

## 2. Bayesian Inference

Some approaches to deriving Bayesian inferences will now be described in detail.

### 2.1. Bayesian Decision Theory

An ingredient that is commonly added to ({f(·|θ):θ∈Θ},π,x) is a loss function, namely, L:Θ×Ψ(Θ)→[0,∞), satisfying $L\left(\theta ,\psi \right)=L\left(\theta^{\prime },\psi \right)$ whenever $\Psi \left(\theta \right)=\Psi \left(\theta^{\prime }\right)$ and L(θ,ψ)=0 only when ψ=Ψ(θ). The goal is to find a procedure, say δ(x)∈Ψ(Θ), which in some sense minimizes the loss L(θ,δ(x)) based on the joint distribution of (θ,x). Given the assumptions on *L*, the loss function can instead be thought of as a map L:Ψ(Θ)×Ψ(Θ)→[0,∞) with $L(\psi ,\psi^{\prime })=0\;$ iff $\psi =\psi^{\prime }$ and the ingredients can be represented as $(\left\{m\left(\cdot \;|\;\psi \right):\psi \in \Psi \left(\Theta \right)\right\},\pi_{\Psi },L,x).$

The goal of a decision analysis is then to find a decision function δ:X→Ψ(Θ) that minimizes the *prior risk*,
$$r\left(\delta \right)=\int_{\Psi (\Theta )}\int_{X}L\left(\psi ,\delta \left(x\right)\right)\;M\left(dx\;|\;\psi \right)\;\Pi_{\Psi }\left(d\psi \right)=\int_{X}r\left(\delta \;|\;x\right)\;M\left(dx\right),$$
where $r\left(\delta \;|\;x\right)=\int_{\Psi (\Theta )}L\left(\psi ,\delta \left(x\right)\right)\;\Pi_{\Psi }\left(d\psi \;|\;x\right)$ is the *posterior risk*. Such a δ is called a Bayes rule and clearly a δ that minimizes r(δ|x) for each *x* is a Bayes rule. Further discussion of the Bayesian decision theory can be found in [12].

As noted in [9], a decision formulation also leads to credible regions for ψ, namely, a γ*-lowest posterior loss-credible region* is defined by
(2)$$D_{\gamma }\left(x\right)=\left\{\psi :r\left(\psi \;|\;x\right)\leq d_{\gamma }\left(x\right)\right\},$$
where $d_{\gamma }\left(x\right)=\inf\{k:\Pi_{\Psi }\left(\left\{\psi :r\left(\psi \;|\;x\right)\leq k\right\}\;|\;x\right)\geq \gamma .$ Note that ψ in (Equation 2) is interpreted as the decision function that takes the value ψ constantly in x. Clearly, as γ→0, set $D_{\gamma }\left(x\right)$ converges to the value of a Bayes rule at x. For example, with quadratic loss, the Bayes rule is given by the posterior mean and a γ-lowest posterior loss region is the smallest sphere centered at the mean containing (at least) γ of the posterior probability.

### 2.2. MAP Inferences

The highest posterior density (HPD) or MAP-based approach to determining inferences constructs credible regions of the following form
(3)$$H_{\gamma }\left(x\right)=\left\{\psi :\pi_{\Psi }\left(\psi \;|\;x\right)\geq h_{\gamma }\left(x\right)\right\},$$
where $\pi_{\Psi }\left(\cdot \;|\;x\right)$ is the marginal posterior density with respect to a support measure $\nu_{\Psi }$ on Ψ(Θ), and $h_{\gamma }\left(x\right)$ is chosen so that $h_{\gamma }\left(x\right)=\sup\left\{k:\Pi_{\Psi }\left(\left\{\psi :\pi_{\Psi }\left(\psi \;|\;x\right)\geq k\right\}\;|\;x\right)\geq \gamma \right\}.$ It follows from (Equation 3) that, to assess the hypothesis $H_{0}:\Psi \left(\theta \right)=\psi_{0},$ we can use the tail probability given by $1-\inf\{\gamma :\psi_{0}\in H_{\gamma }\left(x\right)\}.$ Furthermore, the class of sets $H_{\gamma }\left(x\right)$ is naturally “centered” at the posterior mode (when it exists uniquely) as $H_{\gamma }\left(x\right)$ converges to this point as γ→0. The use of the posterior mode as an estimator is commonly referred to as MAP estimation. We can then think of the size of set $H_{\gamma }\left(x\right),$ say for γ=0.95, as a measure of how accurate the MAP estimator is in a given context. Furthermore, when Ψ(Θ) is an open subset of Euclidean space, then $H_{\gamma }\left(x\right)$ minimizes the volume among all γ-credible regions.

It is well-known that HPD inferences suffer from a defect. In particular, in the continuous case, MAP inferences are not invariant under reparameterization. For example, this means that, if $\psi_{MAP}\left(x\right)$ is the MAP estimate of ψ, then it is not necessarily true that $\Upsilon (\psi_{MAP}\left(x\right))$ is the MAP estimate of τ=Υ(ψ) when Υ is a 1-1 smooth transformation. The non-invariance of a statistical procedure seems very unnatural as it implies that the statistical analysis depends on the parameterization and typically there does not seem to be a good reason for this. Note too that estimates based upon taking posterior expectations will also suffer from this lack of invariance. It is also the case that MAP inferences are not based on a direct characterization of statistical evidence. Both of these issues motivate the development of relative belief inferences.

One justification for MAP inference is decision-theoretic via the loss functions defined in (1). It is common, however, to also consider posterior probabilities of events as expressions of evidence and so think of this approach as evidential in nature. Posterior probabilities, however, express beliefs rather than evidence. For instance, the posterior probability of an event may be very small yet larger than its prior probability, indicating that the data have increased belief in the event’s occurrence. This would suggest that the data provide evidence in favor of the event being true, rather than evidence against it, even though the posterior probability remains small. It appears that evidence is better characterized by how the data change beliefs, rather than by the beliefs themselves.

### 2.3. Relative Belief Inferences

Relative belief inferences, like MAP inferences, are based on the ingredients $(\left\{m\left(\cdot \;|\;\psi \right):\psi \in \Psi \left(\Theta \right)\right\},\pi_{\Psi },x).$ Note that underlying both approaches is the principle (axiom) of conditional probability that says that initial beliefs about ψ, as expressed by the prior $\pi_{\Psi },$ must be replaced by conditional beliefs as expressed by the posterior $\pi_{\Psi }\left(\cdot \;|\;x\right).$ In this approach, however, a measure of statistical evidence is used given by the relative belief ratio,
(4)$$RB_{\Psi }\left(\psi \;|\;x\right)=\frac{\pi_{\Psi }\left(\psi \;|\;x\right)}{\pi_{\Psi }\left(\psi \right)}=\frac{m(x\;|\;\psi )}{m(x)}.$$ The relative belief ratio produces the following conclusions: if $RB_{\Psi }\left(\psi \;|\;x\right)>1,$ then there is evidence in favor of ψ being the true value, if $RB_{\Psi }\left(\psi \;|\;x\right)<1,$ there is evidence against ψ being the true value, and if $RB_{\Psi }\left(\psi \;|\;x\right)=1,$ then there is no evidence either way. These implications follow from a very simple principle of inference.

Principle of evidence: for probability model (Ω,F,P), if C∈F is observed to be true, where P(C)>0, then there is evidence in favor of A∈F being true if P(A|C)>P(A), evidence against A∈F being true if P(A|C)<P(A), and no evidence either way if P(A|C)=P(A).

This principle seems obvious when $\Pi_{\Psi }$ is a discrete probability measure. For the continuous case, where $\Pi_{\Psi }\left(\left\{\psi \right\}\right)=0,$ let $N_{\epsilon }\left(\psi \right)$ be a sequence of neighborhoods of ψ converging nicely to ψ as ϵ→0 (see [13]), then under weak conditions, e.g., $\pi_{\Psi }$ is continuous and positive at ψ,
$$\lim_{\epsilon \to 0}RB_{\Psi }\left(N_{\epsilon }\left(\psi \right)\;|\;x\right)=\lim_{\epsilon \to 0}\frac{\Pi_{\Psi }\left(N_{\epsilon }\left(\psi \right)\;|\;x\right)}{\Pi_{\Psi }\left(N_{\epsilon }\left(\psi \right)\right)}=\frac{\pi_{\Psi }\left(\psi \;|\;x\right)}{\pi_{\Psi }\left(\psi \right)}=RB_{\Psi }\left(\psi \;|\;x\right)$$
and this justifies the general interpretation of $RB_{\Psi }\left(\psi \;|\;x\right)$ as a measure of evidence. The relative belief ratio determines the inferences.

A natural estimate of ψ is the *relative belief estimate*,
$$\psi_{RB}\left(x\right)=\arg\underset{\psi }{\sup}RB_{\Psi }\left(\psi \;|\;x\right)$$
as it has maximum evidence in favor. To assess the accuracy of $\psi_{RB}\left(x\right)$, consider the *plausible region*
$Pl_{\Psi }\left(x\right)=\left\{\psi :RB_{\Psi }\left(\psi \;|\;x\right)>1\right\},$ which is the set of ψ values with evidence supporting them as the true value. The size of $Pl_{\Psi }\left(x\right)$ along with its posterior content $\Pi_{\Psi }\left(Pl_{\Psi }\left(x\right)\;|\;x\right),$ which measures the belief that the true value is in $Pl_{\Psi }\left(x\right),$ provides an assessment of accuracy. So, if $Pl_{\Psi }\left(x\right)$ is “small” and $\Pi_{\Psi }\left(Pl_{\Psi }\left(x\right)\;|\;x\right)\approx 1,$ then $\psi_{RB}\left(x\right)$ is to be considered an accurate estimate of ψ but not otherwise. A relative belief γ-credible region is given by
$$C_{\Psi ,\gamma }\left(x\right)=\left\{\psi :RB_{\Psi }\left(\psi \;|\;x\right)\geq c_{\gamma }\left(x\right)\right\},$$
where $c_{\gamma }\left(x\right)=\sup\left\{c:\Pi_{\Psi }\left(RB_{\Psi }\left(\psi \;|\;x\right)\geq c\;|\;x\right)\geq \gamma \right\},$ for ψ can also be quoted provided $C_{\Psi ,\gamma }\left(x\right)\subset Pl_{\Psi }\left(x\right).$ The containment is necessary, as otherwise, $C_{\Psi ,\gamma }\left(x\right)$ would contain a value ψ for which there is evidence against ψ being the true value.

To assess the hypothesis $H_{0}:\Psi \left(\theta \right)=\psi_{0},$ the value $RB_{\Psi }\left(\psi_{0}\;|\;x\right)$ indicates whether there is evidence in favor of or against $H_{0}.$ The strength of this evidence can be measured by the posterior probability $\Pi_{\Psi }\left(\left\{\psi_{0}\right\}\;|\;x\right),$ as this measures the belief in what the evidence says. So, if $RB_{\Psi }\left(\psi_{0}\;|\;x\right)>1$ and $\Pi_{\Psi }\left(\left\{\psi_{0}\right\}\;|\;x\right)\approx 1,$ then there is strong evidence that $H_{0}$ is true, while when $RB_{\Psi }\left(\psi_{0}\;|\;x\right)<1$ and $\Pi_{\Psi }\left(\left\{\psi_{0}\right\}\;|\;x\right)\approx 0,$ there is strong evidence that $H_{0}$ is false. Since $\Pi_{\Psi }\left(\left\{\psi_{0}\right\}\right)$ can be small, even 0 in the continuous case, it makes more sense to measure the strength of the evidence in such a case by
$$Str_{\Psi }\left(\psi_{0}\;|\;x\right)=\Pi_{\Psi }\left(RB_{\Psi }\left(\psi \;|\;x\right)\leq RB_{\Psi }\left(\psi_{0}\;|\;x\right)\;|\;x\right).$$ If $RB_{\Psi }\left(\psi_{0}\;|\;x\right)>1$ and $Str_{\Psi }\left(\psi_{0}\;|\;x\right)\approx 1,$ then the evidence is strong that $\psi_{0}$ is the true value as there is little belief that the true value of ψ has more evidence in its favor than $\psi_{0}.$ If $RB_{\Psi }\left(\psi_{0}\;|\;x\right)<1$ and $Str_{\Psi }\left(\psi_{0}\;|\;x\right)\approx 0,$ then the evidence is strong that $\psi_{0}$ is not the true value as there is a widespread belief that the true value of ψ has more evidence in its favor than $\psi_{0}.$ There is no reason to quote a single number to measure the strength; both $\Pi_{\Psi }\left(\left\{\psi_{0}\right\}\;|\;x\right)$ and $Str_{\Psi }\left(\psi_{0}\;|\;x\right)$ can be quoted when relevant.

An important aspect of both $Str_{\Psi }\left(\psi_{0}\;|\;x\right)$ and $\Pi_{\Psi }\left(Pl_{\Psi }\left(x\right)\;|\;x\right)$ is what happens as the data increase. To ensure that these behave appropriately, namely, $Str_{\Psi }\left(\psi_{0}\;|\;x\right)\to 0\left(\mathrm{or}1\right)$ when $H_{0}$ is false (or true) and $\Pi_{\Psi }\left(Pl_{\Psi }\left(x\right)\;|\;x\right)\to 1,$ it is necessary to take into account the difference that matters, δ. By this, we mean that there is a distance measure $d_{\Psi }$ on Ψ(Θ)×Ψ(Θ) such that if $d_{\Psi }\left(\psi ,\psi^{\prime }\right)\leq \delta ,$ then in terms of the application, these values are considered equivalent. Such a δ always exists because measurements are always taken to finite accuracy. For example, if ψ is real-valued, then there is a grid of values $\ldots \psi_{-2},\psi_{-1},\psi_{0},\psi_{1},\psi_{2},\ldots$ separated by δ, and inferences are determined using the relative belief ratios of the intervals $[\psi_{i}-\delta /2,\psi_{i}+\delta /2).$ In effect, $H_{0}$ is now $H_{0}:\Psi \left(\theta \right)\in \left[\psi_{0}-\delta /2,\psi_{0}+\delta /2\right).$ When the computations are carried out in this way, then $Str_{\Psi }\left(\psi_{0}\;|\;x\right)$ and $\Pi_{\Psi }\left(Pl_{\Psi }\left(x\right)\;|\;x\right)$ do what is required. As a particular instance of this, see the results in Section 4, where such discretization plays a key role.

It is easy to see that the class of relative-belief credible regions $\left\{C_{\Psi ,\gamma }\left(x\right):\gamma \in \left[0,1\right]\right\}$ for ψ is independent of the marginal prior $\pi_{\Psi }.$ When a value γ∈[0,1] is specified, however, set $C_{\Psi ,\gamma }\left(x\right)$ depends on $\pi_{\Psi }$ through $c_{\gamma }\left(x\right).$ So, the form of relative belief inferences about ψ is completely robust to the choice of $\pi_{\Psi }$ but the quantification of the uncertainty in the inferences is not. For example, when ψ=Ψ(θ)=θ, then $\theta_{RB}\left(x\right)$ is the MLE; however, in general, $\psi_{RB}\left(x\right)$ is the maximizer of the integrated likelihood m(x|ψ). Similarly, relative belief regions are likelihood regions in cases of the full parameter and integrated likelihood regions. As such, likelihood regions can be seen as essentially Bayesian in character with a clear and precise characterization of evidence through the relative belief ratio, and now have probability assignments through the posterior. A relative belief ratio, $RB_{\Psi }\left(\psi \;|\;x\right),$ while proportional to an integrated likelihood, cannot be multiplied by an arbitrary positive constant—as with a likelihood—without losing its interpretation in measuring statistical evidence. It has been established in [14] that relative belief inferences for ψ are optimally robust to the prior $\pi_{\Psi }.$

As can be seen from (Equation 4), relative belief inferences are always invariant under smooth reparameterizations; this is at least one reason why they are preferable to MAP inferences. Any rule for measuring evidence, which satisfies the principle of evidence, also produces valid estimates, as these lie in $Pl_{\Psi }\left(x\right)$ and so will have the same “accuracy” as $\psi_{RB}\left(x\right).$ For example, if instead of the relative belief ratio, the difference $\pi_{\Psi }\left(\psi \;|\;x\right)-\pi_{\Psi }\left(\psi \right)$ is used as the measure of evidence with a cut-off of 0, then this satisfies the principle of evidence, but the estimate is no longer necessarily invariant under reparameterizations. The Bayes factor with a cut-off of 1 is also a valid measure of evidence but there are a number of reasons why the relative belief ratio is to be preferred to the Bayes factor for general inferences (see [15]).

We will now consider a simple example that illustrates the various concepts just discussed.

**Example 1.** 
*Location normal*

*Suppose that we have a sample $x=(x_{1},\ldots ,x_{n})$ from a $N(\theta ,\sigma_{0}^{2})$ where the mean $\theta \in \Theta =R^{1}$ is unknown and the variance $\sigma_{0}^{2}$ is assumed known. Suppose interest lies in making inferences about ψ=Ψ(θ)=θ, and the prior π on θ is given by a $N(\theta_{0},\tau_{0}^{2})$ distribution. In this context, $\bar{x}$ serves as a minimal sufficient statistic, allowing the focus to be restricted to the $N(\theta ,\sigma_{0}^{2}/n)$ model while ignoring the remaining aspects of the data, at least for inference. Certainly the residuals $\left(x_{1}-\bar{x},\ldots ,x_{n}-\bar{x}\right)$ are relevant for model checking. The prior predictive density m of $\bar{x}$ is then given by the density of a $N(\theta_{0},\tau_{0}^{2}+\sigma_{0}^{2}/n)$ distribution and, as discussed in [7], this is relevant for checking for prior-data conflicts via the tail probability $M(m\left(\bar{X}\right)\leq m\left(\bar{x}\right))$ with small values indicating the existence of a conflict.*

*The posterior Π(·|x) of θ is given by*

$$\theta \;|\;\bar{x}∼N\left({\left(n/\sigma_{0}^{2}+1/\tau_{0}^{2}\right)}^{-1}\left(n\bar{x}/\sigma_{0}^{2}+\theta_{0}/\tau_{0}^{2}\right),{\left(n/\sigma_{0}^{2}+1/\tau_{0}^{2}\right)}^{-1}\right).$$

*As such, $\theta_{MAP}\left(\bar{x}\right)={\left(n/\sigma_{0}^{2}+1/\tau_{0}^{2}\right)}^{-1}\left(n\bar{x}/\sigma_{0}^{2}+\theta_{0}/\tau_{0}^{2}\right).$ If, as is common, squared error loss is employed, then the Bayes rule for estimating θ is given by $\theta_{MAP}\left(\bar{x}\right),$ as this is also the posterior mean. On the other hand, the relative belief ratio is given by*

$$\begin{array}{cc} RB(\theta \;|\;\bar{x}) & =\frac{\pi (\theta \;|\;\bar{x})}{\pi (\theta )}=\frac{m(\bar{x}\;|\;\theta )}{m(\theta )}\\ \; & ={\left(\frac{\tau_{0}^{2}+\sigma_{0}^{2}/n}{\sigma_{0}^{2}/n}\right)}^{1/2}\frac{\exp(-n{\left(\bar{x}-\theta \right)}^{2}/2\sigma_{0}^{2})}{\exp(-{\left(\tau_{0}^{2}+\sigma_{0}^{2}/n\right)}^{-1}{\left(\bar{x}-\theta_{0}\right)}^{2}/2)} \end{array}$$

*as, since there are no nuisance parameters, m(·|θ) equals the sampling density of $\bar{x}.$*

*From this, it is immediate that $\theta_{RB}\left(\bar{x}\right)=\bar{x}$, which is the MLE, a result that is generally true for relative belief when estimating the full model parameter. The plausible interval for θ is then, putting $r_{0}=\left(\sigma_{0}^{2}/n\right)/\left(\tau_{0}^{2}+\sigma_{0}^{2}/n\right),$ given by*

$$Pl\left(\bar{x}\right)=\left\{\theta :RB\left(\theta \;|\;\bar{x}\right)>1\right\}=\left(\bar{x}\pm \sigma_{0}\sqrt{\frac{{\left(\tau_{0}^{2}+\sigma_{0}^{2}/n\right)}^{-1}{\left(\bar{x}-\theta_{0}\right)}^{2}-\log\left(r_{0}\right)}{n}}\right)$$

*and note that $-\log(r_{0})>0$, so this interval is always defined. The length of $Pl(\bar{x})$ and its posterior content, computed using Π(·|x), provide a measure of the accuracy of $\theta_{RB}\left(\bar{x}\right).$ Notice that $Pl(\bar{x})$ converges almost surely to the degenerate interval consisting of the true value of θ and the posterior content of the interval converges to 1 as n→∞.*

*To assess a hypothesis, say $H_{0}:\theta =\theta_{0},$ the relevant relative belief ratio is as follows:*

$$RB\left(\theta_{0}\;|\;\bar{x}\right)=r_{0}^{1/2}\exp\left(-\frac{r_{0}-1}{r_{0}}\frac{n}{\sigma_{0}^{2}}\frac{{\left(\bar{x}-\theta_{0}\right)}^{2}}{2}\right).$$

*This gives evidence in favor (or against) $H_{0}$ when*

$$\frac{n{\left(\bar{x}-\theta_{0}\right)}^{2}}{\sigma_{0}^{2}}<\left(>\right)\frac{r_{0}\log r_{0}}{r_{0}-1}$$

*and note that the right-hand side is always positive. The strength of this evidence is given by*

$$\begin{array}{cc} \; & Str\left(\theta_{0}\;|\;x\right)=\Pi \left(RB\left(\theta \;|\;x\right)\leq RB\left(\theta_{0}\;|\;x\right)\;|\;x\right)=\Pi (|\bar{x}-\theta |\geq |\bar{x}-\theta_{0}\left|\;|\;x\right)\\ \; & =F(\bar{x}+|\bar{x}-\theta_{0}|;{\left(n/\sigma_{0}^{2}+1/\tau_{0}^{2}\right)}^{-1}\left(n\bar{x}/\sigma_{0}^{2}+\theta_{0}/\tau_{0}^{2}\right),{\left(n/\sigma_{0}^{2}+1/\tau_{0}^{2}\right)}^{-1})-\\ \; & F(\bar{x}-|\bar{x}-\theta_{0}|;{\left(n/\sigma_{0}^{2}+1/\tau_{0}^{2}\right)}^{-1}\left(n\bar{x}/\sigma_{0}^{2}+\theta_{0}/\tau_{0}^{2}\right),{\left(n/\sigma_{0}^{2}+1/\tau_{0}^{2}\right)}^{-1}), \end{array}$$

*where F(·;μ,λ) denotes the N(μ,λ) cdf. As n→∞, the strength converges to 0 (the strongest possible evidence against) when $H_{0}$ is false and converges to 1 (the strongest possible evidence in favor) when $H_{0}$ is true.*


## 3. Estimation: Discrete Parameter Space

The following theorem presents the basic definition of the loss function for the parameter of interest ψ=Ψ(θ) when the set of possible values of ψ, namely, Ψ(Θ)={ψ:ψ=Ψ(θ)forsomeθ∈Θ}, is finite. This establishes an important optimality result. The indicator function for the set *A* is denoted as $I_{A}$.

**Theorem 1.** 
*Suppose that $\pi_{\Psi }\left(\psi \right)>0$ for every ψ∈Ψ(Θ), Ψ(Θ) is finite with $\nu_{\Psi }$ equal to counting measure on Ψ(Θ). Then for the loss function*

(5)
$$L_{RB}\left(\theta ,\psi \right)=\frac{I_{{\left\{\psi \right\}}^{c}}\left(\Psi \left(\theta \right)\right)}{\pi_{\Psi }\left(\Psi \left(\theta \right)\right)},$$

*the relative belief estimator $\psi_{RB}$ is a Bayes rule.*


**Proof.** We have that
(6)$$\begin{array}{cc} r(\delta \;|\;x) & =\int_{\Psi (\Theta )}\frac{I_{{\left\{\delta \left(x\right)\right\}}^{c}}\left(\psi \right)}{\pi_{\Psi }\left(\psi \right)}\;\Pi_{\Psi }\left(d\psi \;|\;x\right)\\ \; & =\int_{\Psi (\Theta )}RB_{\Psi }\left(\psi \;|\;x\right)\;\nu_{\Psi }\left(d\psi \right)-RB_{\Psi }\left(\delta \left(x\right)\;|\;x\right). \end{array}$$ Since Ψ(Θ) is finite, the first term in (Equation 6) is finite and a Bayes rule at *x* is given by the value δ(x) that maximizes the second term. Therefore, $\psi_{RB}$ is a Bayes rule. □

The loss function $L_{RB}$ seems very natural. Beliefs about the true value of ψ are expressed by the prior $\pi_{\Psi }$. As such, consider values where $\pi_{\Psi }\left(\psi \right)$ is very low and ψ is indeed false. It would then be misleading if inferences suggested such a value as being true. So it is appropriate for such values to bear large losses. In a sense, the statistician is acknowledging what such values are by the choice of the prior. Of course, the prior may be wrong in the sense that the bulk of its mass is placed in a region where the true value of ψ does not lie. This is why checking for prior-data conflict, before conducting inference, is always recommended. Procedures for checking priors were discussed in [16,17], and an approach to replacing priors found to be at fault was developed in [7]. The loss function $L_{RB}$ motivates the other losses for relative belief discussed here, making this comment relevant to those losses as well.

The prior risk of δ satisfies the following,
(7)$$r\left(\delta \right)=\int_{\Psi (\Theta )}\int_{X}\frac{I_{{\left\{\delta \left(x\right)\right\}}^{c}}\left(\psi \right)}{\pi_{\Psi }\left(\psi \right)}\;M\left(dx\;|\;\psi \right)\;\Pi_{\Psi }\left(d\psi \right)=\sum_{\psi \in \Psi (\Theta )}M\left(\delta \left(x\right)\neq \psi \;|\;\psi \right),$$
where M(·|ψ) is the conditional prior predictive probability measure of the data given ψ, so (Equation 7) is the sum of the conditional prior error probabilities over all ψ values. If instead the loss function is taken to be $L_{MAP}\left(\theta ,\psi \right)=I_{{\left\{\psi \right\}}^{c}}\left(\Psi \left(\theta \right)\right),$ as in (Equation 1), then the same proof used in Theorem 1 establishes that $\psi_{MAP}$ is a Bayes rule with respect to this loss, and the prior risk is given by,
(8)$$\sum_{\psi }M\left(\delta \left(x\right)\neq \psi \;|\;\psi \right)\pi_{\Psi }\left(\psi \right),$$
which represents the prior probability of making an error. Both $L_{MAP}$ and $L_{RB}$ are two-valued loss functions but, when an incorrect decision is made, the loss is constant in Ψ(θ) for $L_{MAP}$, while it equals the reciprocal of the prior probability of Ψ(θ) for $L_{RB}$. So, $L_{RB}$ penalizes an incorrect decision much more severely when the true value of Ψ(θ) is in the tails of the prior. Note that $\psi_{MAP}=\psi_{RB}$ when $\Pi_{\Psi }$ is uniform. It is evident that (Equation 7) serves as an upper bound to (Equation 8), indicating that controlling losses based on $L_{RB}$ automatically controls the losses based on $L_{MAP}.$

As already noted, $RB_{\Psi }\left(\psi \;|\;x\right)$ is proportional to the integrated likelihood of ψ. So, under the conditions of Theorem 1, the maximum integrated likelihood estimator is a Bayes rule. Furthermore, the Bayes rule is the same for every choice of $\pi_{\Psi }$ and only depends on the full prior through the conditional prior Π(·|ψ) placed on the nuisance parameters. When ψ=θ, then $\theta_{RB}\left(x\right)$ is the MLE of θ and so the MLE of θ is a Bayes rule for every prior π.

Note that when $\Psi \left(\Theta \right)=\left\{\psi_{0},\psi_{1}\right\}$, then $RB_{\Psi }\left(\psi_{0}\;|\;x\right)>\left(<\right)1$ iff $RB_{\Psi }\left(\psi_{1}\;|\;x\right)<\left(>\right)1$, so $\psi_{RB}\left(x\right)=\psi_{0}$ when $RB_{\Psi }\left(\psi_{0}\;|\;x\right)>1$, and $\psi_{RB}\left(x\right)=\psi_{1}$ otherwise. This is the classical context for hypothesis testing, where $\psi_{RB}\left(x\right)=\psi_{0}$ can be viewed as acceptance of the hypothesis $H_{0}:\theta \in \Psi^{-1}\left\{\psi_{0}\right\}$, and $\psi_{RB}\left(x\right)=\psi_{1}$ as rejection of $H_{0}.$ Theorem 1 establishes that relative belief offers a Bayes rule for the hypothesis testing problem.

The loss function (Equation 5) does not provide meaningful results when Ψ(Θ) is infinite as (Equation 7) shows that r(δ) will be infinite. So, we modify (Equation 5) via a parameter η>0 and define the loss function as follows,
(9)$$L_{RB,\eta }\left(\theta ,\psi \right)=\frac{I_{{\left\{\psi \right\}}^{c}}\left(\Psi \left(\theta \right)\right)}{\max(\eta ,\pi_{\Psi }\left(\Psi \left(\theta \right)\right))}.$$ Note that $L_{RB,\eta }$ is bounded by 1/η. This loss function is like (Equation 5) but does not allow for arbitrarily large losses. The following result shows that we can restrict attention to values of η that are sufficiently small.

**Theorem 2.** 
*Suppose that $\pi_{\Psi }\left(\psi \right)>0$ for every ψ∈Ψ(Θ), where Ψ(Θ) is countable with $\nu_{\Psi }$ equal to counting measure, and that $\psi_{RB}\left(x\right)$ is the unique maximizer of $RB_{\Psi }\left(\psi \;|\;x\right)$ for all x. For the loss function (Equation 9) and Bayes rule $\delta_{\eta },$ then $\delta_{\eta }\left(x\right)\to \psi_{RB}\left(x\right)$ as η→0, for every x∈X.*


The proof of Theorem 2 also establishes the following result.

**Corollary 1.** 
*For all sufficiently small η, the value of a Bayes rule at x is given by $\psi_{RB}\left(x\right).$*


The following is an immediate consequence of Theorem 1 and Corollary 1 as $\psi_{RB}$ is a Bayes rule.

**Corollary 2.** 
*$\psi_{RB}$ is an admissible estimator with respect to the loss function $L_{RB}$ when Ψ(Θ) is finite, and with respect to loss $L_{RB,\eta },$ when η is sufficiently small, and Ψ(Θ) is countable.*


In a general estimation problem, δ is risk-unbiased with respect to a loss function *L* if $E_{\theta }\left(L\left(\theta^{\prime },\delta \left(x\right)\right)\right)\geq E_{\theta }\left(L\left(\theta ,\delta \left(x\right)\right)\right)$ for all $\theta^{\prime },\theta \in \Theta .$ This says that, on average, δ(x) is closer to the true value than any other value when we interpret L(θ,δ(x)) as a measure of distance between δ(x) and Ψ(θ). A definition of *Bayesian-unbiasedness* for δ with respect to *L* is given by the inequality,
$$\int_{\Theta }\int_{\Theta }E_{\theta }\left(L\left(\theta^{\prime },\delta \left(x\right)\right)\right)\;\Pi \left(d\theta \right)\;\Pi \left(d\theta^{\prime }\right)\geq \int_{\Theta }E_{\theta }\left(L\left(\theta ,\delta \left(x\right)\right)\right)\;\Pi \left(d\theta \right)=r\left(\delta \right),$$
as this retains the idea of being closer, on average, to the true value than a false value. We will now consider a family of loss functions defined as follows,
(10)$$L\left(\theta ,\psi \right)=I_{{\left\{\psi \right\}}^{c}}\left(\Psi \left(\theta \right)\right)h\left(\Psi \left(\theta \right)\right)$$
where *h* is a nonnegative function satisfying $\int_{\Theta }h\left(\Psi \left(\theta \right)\right)\;\Pi \left(d\theta \right)<\infty \;$. This includes $L_{RB}$ and $L_{MAP}$ when Ψ(Θ) is finite and $L_{RB,\eta }.$

**Theorem 3.** 
*If Ψ(Θ) is finite or countable, then $\psi_{RB}\left(x\right)$ is Bayesian-unbiased under the loss function (Equation 10).*


Suppose after observing *x*, there is a need to predict a future (or concealed) value y∈Y, where $y∼g_{\delta (\theta )}\left(y\;|\;x\right),$ a density with respect to the support measure $\mu_{Y}$ on Y, and it is assumed that the true value of θ in the model for *x*, gives the true value of δ(θ). The prior predictive density of *y* is given by $q\left(y\right)=\int_{\Theta }\int_{X}\pi \left(\theta \right)f_{\theta }\left(x\right)g_{\delta (\theta )}\left(y\;|\;x\right)\;\mu \left(dx\right)\;\nu \left(d\theta \right)$ while the posterior predictive density is $q\left(y\;|\;x\right)=\int_{\Theta }\pi \left(\theta \;|\;x\right)g_{\delta (\theta )}\left(y\;|\;x\right)\;\nu \left(d\theta \right).$ The relative belief ratio for a future value of *y* is, thus, $RB_{Y}\left(y\;|\;x\right)=q\left(y\;|\;x\right)/q\left(y\right)$, and the relative belief prediction, namely, the value that maximizes $RB_{Y}\left(\cdot \;|\;x\right),$ is denoted as $y_{RB}\left(x\right).$ When Y is finite, using the same argument as in Theorem 1, $y_{RB}$ is a Bayes rule under the loss function $L_{RB}\left(y,y^{\prime }\right)=I_{{\left\{y\right\}}^{c}}\left(y^{\prime }\right)/q\left(y\right).$ Also, it can be demonstrated that $y_{RB}$ is a limit of the Bayes rule when Y is countable.

We will now consider a common application where Ψ(Θ) is finite.

**Example 2.** 
*Classification*
*For a classification problem, there are k categories $\{\psi_{1},\ldots ,\psi_{k}\},$ prescribed by a function Ψ, where $\pi_{\Psi }\left(\psi_{i}\right)>0$ for each i. Estimating ψ is then equivalent to classifying the data as having come from one of the distributions in the classes specified by $\Psi^{-1}\left\{\psi_{i}\right\}.$ The standard Bayesian solution to this problem is to use $\psi_{MAP}\left(x\right)$ as the classifier. From (Equation 8), we have that $\psi_{MAP}\left(x\right)$ minimizes the prior probability of misclassification, while from (Equation 7), $\psi_{RB}\left(x\right)$ minimizes the sum of the probabilities of misclassification. The essence of the difference is that $\psi_{RB}\left(x\right)$ treats the misclassification errors equally while $\psi_{MAP}\left(x\right)$ weights the errors by their prior probabilities*.
*The following shows that minimizing the sum of error probabilities is often more appropriate than minimizing the weighted sum. Suppose that k=2 and x∼ Bernoulli$\left(\psi_{0}\right)$ or x∼ Bernoulli$\left(\psi_{1}\right)$ with $\pi (\psi_{0})=1-\epsilon$ and $\pi (\psi_{1})=\epsilon$ representing the known proportions of individuals coming from population 0 or 1. For example, consider $\psi_{0}$ as the probability of a positive diagnostic test for a disease in the non-diseased population while $\psi_{1}$ is this probability for the diseased population. Suppose that $\psi_{0}/\psi_{1}$ is very small, indicating that the test is successful at identifying the disease while not yielding many false positives and that ϵ is very small, so the disease is rare. The challenge then becomes assigning a randomly chosen individual to a population based on their test results.*
*The posterior is given by $\pi \left(\psi_{0}\;|\;1\right)=\psi_{0}\left(1-\epsilon \right)/\left(\psi_{0}\left(1-\epsilon \right)+\psi_{1}\epsilon \right)$ and $\pi \left(\psi_{0}\;|\;0\right)=\left(1-\psi_{0}\right)\left(1-\epsilon \right)/\left(\left(1-\psi_{0}\right)\left(1-\epsilon \right)+\left(1-\psi_{1}\right)\epsilon \right).$ Therefore,*$$\begin{array}{cc} \psi_{MAP}\left(1\right) & =\left\{\begin{array}{cc} \psi_{0} & if\psi_{0}/\psi_{1}>\epsilon /\left(1-\epsilon \right)\\ \psi_{1} & otherwise \end{array}\;\right.\\ \psi_{MAP}\left(0\right) & =\left\{\begin{array}{cc} \psi_{0} & if\left(1-\psi_{0}\right)/\left(1-\psi_{1}\right)>\epsilon /\left(1-\epsilon \right)\\ \psi_{1} & otherwise \end{array}\;\right. \end{array}$$*This implies that $\psi_{MAP}$ will always classify a person to the non-diseased population when ϵ is small enough, e.g., when $\psi_{0}=0.05,\psi_{1}=0.80,$ and ϵ<0.0625. In contrast, in this situation, $\psi_{RB}$ always classifies an individual with a positive test to the diseased population and the non-diseased population for a negative test. Since $M(\cdot \;|\;\psi_{i})$ is the Bernoulli$\left(\psi_{i}\right)$ distribution, when $\psi_{0}<\psi_{1}$ and ϵ are small enough, we have the following:*$$\begin{array}{cc} M(\psi_{MAP} & \neq \psi_{0}\;\left|\;\psi_{0}\right)+M\left(\psi_{MAP}\neq \psi_{1}\;|\;\psi_{1}\right)=0+1=1,\\ M(\psi_{RB} & \neq \psi_{0}\;\left|\;\psi_{0}\right)+M\left(\psi_{RB}\neq \psi_{1}\;|\;\psi_{1}\right)=\psi_{0}+\left(1-\psi_{1}\right)=0.25. \end{array}$$*This clearly illustrates the difference between these two procedures as $\psi_{RB}$ does better than $\psi_{MAP}$ on the diseased population when $\psi_{0}$ is small and $\psi_{1}$ is large, as would be the case for a good diagnostic. Of course, $\psi_{MAP}$ minimizes the overall error rate, but at the price of ignoring the most important class in this problem, namely, those who have the disease. Note that this example can be extended to the situation where we need to estimate the $\psi_{i}$ based on samples from the respective populations, but this will not materially affect the overall conclusions*.
*We will now consider a situation where (x,c) is such that $x\;|\;c∼f_{c},c\;|\;\epsilon ∼$ Bernoulli(ϵ), where $f_{0}$ and $f_{1}$ are known but ϵ is unknown with a prior π. This is a generalization of the previous discussion, where ϵ is assumed to be known. Then, based on a sample $\left(x_{1},c_{1}\right),\ldots ,\left(x_{n},c_{n}\right)$ from the joint distribution, the goal is to predict the value $c_{n+1}$ for a newly observed $x_{n+1}.$*

*The prior of c is $q\left(c\right)=\int_{0}^{1}{\left(1-\epsilon \right)}^{1-c}\epsilon^{c}\pi \left(\epsilon \right)\;d\epsilon \;$, and if ϵ∼ beta(α,β), the prior predictive of $c_{n+1}$ is Bernoulli(α/(α+β)). The posterior predictive density of $c_{n+1}$ equals, where $\bar{c}=n^{-1}\sum_{i=1}^{n}c_{i},$*

$$\begin{array}{cc} \; & q(c\;|\;\left(x_{1},c_{1}\right),\ldots ,\left(x_{n},c_{n}\right),x_{n+1})\\ \; & \propto {\left(f_{0}\left(x_{n+1}\right)\right)}^{1-c}{\left(f_{1}\left(x_{n+1}\right)\right)}^{c}\int_{0}^{1}\epsilon^{n\bar{c}+c}{\left(1-\epsilon \right)}^{n\left(1-\bar{c}\right)+\left(1-c\right)}\pi \left(\epsilon \right)\;d\epsilon \\ \; & =f_{c}\left(x_{n+1}\right)\Gamma \left(\alpha +n\bar{c}+c\right)\Gamma \left(\beta +n\left(1-\bar{c}\right)+1-c\right). \end{array}$$

*It follows that, suppressing the dependence on the data, we have the following,*

(11)
$$\begin{array}{cc} c_{MAP} & =\left\{\begin{array}{cc} 1 & if\;\frac{f_{1}\left(x_{n+1}\right)}{f_{0}\left(x_{n+1}\right)}\frac{\left(\alpha +n\bar{c}\right)}{\left(\beta +n(1-\bar{c})\right)}>1\\ 0 & otherwise, \end{array}\right.\\ c_{RB} & =\left\{\begin{array}{cc} 1 & if\;\frac{f_{1}\left(x_{n+1}\right)}{f_{0}\left(x_{n+1}\right)}\frac{\beta \left(\alpha +n\bar{c}\right)}{\alpha \left(\beta +n(1-\bar{c})\right)}<1\\ 0 & otherwise \end{array}\right. \end{array}$$

*Note that $c_{MAP}$ and $c_{RB}$ are identical whenever α=β.*

*From these formulas, it is apparent that a substantial difference will arise between $c_{MAP}$ and $c_{RB}$ when either α or β is much bigger than the other. As in Example 2, these correspond to situations where we believe that ϵ or 1−ϵ is very small. Suppose we take α=1 and let β be relatively large, as this corresponds to knowing a priori that ϵ is very small. Then (Equation 11) implies that $c_{MAP}\leq c_{RB}$ and so $c_{RB}=1$ whenever $c_{MAP}=1.$ A similar conclusion arises when we take β=1 and α<1.*

*To see what kind of improvement is possible, we consider a simulation study. Let $f_{0}$ be a N(0,1) density, $f_{1}$ be a N(μ,1) density, and n=10 and the prior on ϵ be beta(1,β). Table 1 presents the Bayes risks for $c_{MAP}\;$ and $c_{RB}$ for various choices of β when μ=1. When β=1, they are equivalent, but we see that as β rises, the performance of $c_{MAP}$ deteriorates while $c_{RB}$ improves. Large values of β correspond to having information where ϵ is small. When β=14, about 0.50 of the prior probability is to the left of 0.05; with β=32, about 0.80 of the prior probability is to the left of 0.05; and with β=100, about 0.99 of the prior probability is to the left of 0.05. We see that the misclassification rates for the small group (c=1) stay about the same for $c_{RB}$ as β increases while they deteriorate markedly for $c_{MAP}$ as the MAP procedure basically ignores the small group.*

*We also investigated other choices for n and μ. There is very little change as n increases. When μ moves toward 0 and μ moves away from 0, the error rates go up and go down, as one would expect; $c_{RB}$ always dominates $c_{MAP}.$*


## 4. Estimation: Continuous Parameter Space

When ψ has a continuous prior distribution, the argument in Theorem 2 does not work, as $\Pi_{\Psi }\left(\left\{\delta \left(x\right)\right\}\;|\;x\right)=0.$ There are several possible ways to proceed but one approach is to use a discretization of the problem that uses Theorem 2. For this, we will assume that the spaces involved are locally Euclidean, the mappings are sufficiently smooth, and take the support measures to be the analogs of Euclidean volume on the respective spaces. While the argument presented is broadly applicable, it has been simplified in this context by assuming that all spaces are open subsets of Euclidean spaces, with the support measures being the Euclidean volume on these sets.

For each λ>0, suppose there is a discretization $\left\{B_{\lambda }\left(\psi \right):\psi \in \Psi \left(\Theta \right)\right\}$ of Ψ(Θ) into a countable number of subsets with the following properties: $\psi \in B_{\lambda }\left(\psi \right),\Pi_{\Psi }\left(B_{\lambda }\left(\psi \right)\right)>0$ and $\sup_{\psi \in \Psi }$diam$(B_{\lambda }\left(\psi \right))\to 0$ as λ→0. So, if $\psi^{\prime }\in B_{\lambda }\left(\psi \right),$ then $B_{\lambda }\left(\psi^{\prime }\right)=B_{\lambda }\left(\psi \right).$ For example, $B_{\lambda }\left(\psi \right)$ could be equal volume rectangles in $R^{k}.$ Further, we assume that $\Pi_{\Psi }\left(B_{\lambda }\left(\psi \right)\right)/\nu_{\Psi }\left(B_{\lambda }\left(\psi \right)\right)\to \pi_{\Psi }\left(\psi \right)$ as λ→0 for every ψ. This will hold whenever $\pi_{\Psi }$ is continuous everywhere and $B_{\lambda }\left(\psi \right)$ converges nicely to {ψ} as λ→0. Let $\psi_{\lambda }\left(\psi \right)$ denote a point in $B_{\lambda }\left(\psi \right)$ such that $\psi_{\lambda }\left(\psi \right)=\psi_{\lambda }\left(\psi^{\prime }\right)$ whenever $\psi ,\psi^{\prime }\in \in B_{\lambda }\left(\psi \right)$ and put $\Psi_{\lambda }=\left\{\psi_{\lambda }\left(\psi \right):\psi \in \Psi \left(\Theta \right)\right\}.$ So, $\Psi_{\lambda }$ is a discretized version of Ψ(Θ). We will call this a *regular discretization* of Ψ(Θ). The discretized prior on $\Psi_{\lambda }$ is $\pi_{\Psi ,\lambda }\left(\psi_{\lambda }\left(\psi \right)\right)=\Pi_{\Psi }\left(B_{\lambda }\left(\psi \right)\right)$ and the discretized posterior is $\pi_{\Psi ,\lambda }\left(\psi_{\lambda }\left(\psi \right)\;|\;x\right)=\Pi_{\Psi }\left(B_{\lambda }\left(\psi \right)\;|\;x\right).$

The loss function for the discretized problem is defined in Theorem 2 as follows,
(12)$$L_{RB,\lambda ,\eta }\left(\theta ,\psi_{\lambda }\left(\psi \right)\right)=\frac{I_{\left\{\psi_{\lambda }\left(\psi \right)\right\}}\left(\psi_{\lambda }\left(\Psi \left(\theta \right)\right)\right)}{\max(\eta ,\pi_{\Psi ,\lambda }\left(\psi_{\lambda }\left(\Psi \left(\theta \right)\right)\right))}$$
and let $\delta_{\lambda ,\eta }\left(x\right)$ denote a Bayes rule for this problem.

**Theorem 4.** 
*Suppose that $\pi_{\Psi }$ is positive and continuous and we have a regular discretization of Ψ. Furthermore, suppose that $\psi_{RB}\left(x\right)$ is the unique maximizer of $RB_{\Psi }\left(\psi \;|\;x\right)$ and for any ϵ>0,*

$$\underset{\left\{\psi :||\psi -\psi_{RB}\left(x\right)||\geq \epsilon \right\}}{\sup}RB_{\Psi }\left(\psi \;|\;x\right)<RB_{\Psi }\left(\psi_{RB}\left(x\right)\;|\;x\right).$$

*Then, there exists η(λ)↓0 as λ→0 such that a Bayes rule $\delta_{\lambda ,\eta (\lambda )}\left(x\right),$ under the loss $L_{RB,\lambda ,\eta (\lambda )},$ converges to $\psi_{RB}\left(x\right)$ as λ→0 for all x.*


Theorem 4 states that $\psi_{RB}\left(x\right)$ is a limit of Bayes rules. So, when Ψ(θ)=θ, we have the result that the MLE is a limit of the Bayes rule, and more generally, the MLE from an integrated likelihood is a limit of Bayes rules. The regularity conditions stated in Theorem 4 hold in many common statistical problems.

Now let ${\hat{\psi }}_{\lambda }\left(x\right)$ be the relative belief estimate from the discretized problem, i.e., ${\hat{\psi }}_{\lambda }\left(x\right)$ maximizes $RB_{\Psi }\left(B_{\lambda }\left(\psi \right)\;|\;x\right)$ as a function of $\psi \in \Psi_{\lambda }.$ The following is immediate from the proof of Theorem 4, Theorem 3, and Corollary 2:

**Corollary 3.** 
*${\hat{\psi }}_{\lambda }$ is admissible and Bayesian-unbiased for the discretized problem, and ${\hat{\psi }}_{\lambda }\left(x\right)\to \psi_{RB}\left(x\right)$ as λ→0 for every x.*


By similar arguments, an analog of Theorem 4 for $\psi_{MAP}$ can be established. In this case, a simpler development can be followed in certain situations by using the loss function $I_{B_{\lambda }^{c}\left(\psi \right)}\left(\Psi \left(\theta \right)\right)$. For this, the posterior risk of δ in the discretized problem, is given by $1-\Pi_{\Psi }\left(B_{\lambda }\left(\delta \left(x\right)\right)\;|\;x\right)=1-\pi_{\Psi }\left(\delta^{\prime }\left(x\right)\;|\;x\right)\nu_{\Psi }\left(B_{\lambda }\left(\delta \left(x\right)\right)\right)$ for some $\delta^{\prime }\left(x\right)\in B_{\lambda }\left(\delta \left(x\right)\right).$ Now suppose $B_{\lambda }\left(\psi \right)$ is a cube centered at ψ of edge length δ. Suppose that for each ϵ>0, there exists λ(ϵ)>0, such that, when $||\psi -\psi_{MAP}\left(x\right)\left|\right|>\lambda \left(\epsilon \right)$, then
$$\;pi_{\Psi }\left(\psi \;|\;x\right)<\underset{\psi^{\prime }\in B_{\lambda (\epsilon )}\left(\psi_{MAP}\left(x\right)\right)}{\inf}\pi_{\Psi }\left(\psi^{\prime }\;|\;x\right).$$ Since $\nu_{\Psi }\left(B_{\lambda }\left(\psi \right)\right)$ is constant, a Bayes rule $\delta_{\lambda (\epsilon )}$ must then satisfy $\left|\right|\delta_{\lambda (\epsilon )}\left(x\right)-\psi_{MAP}\left(x\right)\left|\right|<\epsilon$. This proves that $\psi_{MAP}$ is a limit of the Bayes rules. By contrast, for the loss
$$I_{B_{\lambda }^{c}\left(\psi \right)}\left(\Psi \left(\theta \right)\right)/\Pi_{\Psi }\left(B_{\lambda }\left(\Psi \left(\theta \right)\right)\right),$$
the posterior risk of δ is given by,
$$\int_{\Psi }{\left\{\Pi_{\Psi }\left(B_{\lambda }\left(\psi \right)\right)\right\}}^{-1}\Pi_{\Psi }\left(d\psi \;|\;x\right)-\int_{B_{\lambda }\left(\delta \left(x\right)\right)}{\left\{\Pi_{\Psi }\left(B_{\lambda }\left(\psi \right)\right)\right\}}^{-1}\Pi_{\Psi }\left(d\psi \;|\;x\right)$$
and the first term is generally unbounded unless Ψ(Θ) is compact.

We will now consider an important example.

**Example 3.** 
*Regression*

*Suppose that y=Xβ+e, where $y\in R^{n},X\in R^{n\times k}$ is fixed of rank $k,\beta \in R^{n\times k},$ and $e∼N_{n}\left(0,\sigma^{2}I\right).$ To simplify the discussion, we will assume that $\sigma^{2}$ is known but this is not necessary. Let π be a prior density for β. For every π, having observed (X,y), then $\beta_{RB}\left(y\right)=b={\left(X^{\prime }X\right)}^{-1}X^{\prime }y$, the MLE of β.*

*It is interesting to contrast this result with more standard Bayesian estimates such as MAP or the posterior mean. For example, suppose that $\beta ∼N_{k}\left(0,\tau^{2}I\right).$ Then the posterior distribution of β is $N_{k}\left(\beta_{post}\left(y\right),\Sigma_{post}\right)$, where*

$$\beta_{post}\left(y\right)=\Sigma_{post}\left(\sigma^{-2}X^{\prime }Xb\right),\;\Sigma_{post}={\left(\tau^{-2}I+\sigma^{-2}X^{\prime }X\right)}^{-1}$$

*and note that $\beta_{MAP}\left(y\right)=\beta_{post}\left(y\right).$ Writing the spectral decomposition of $X^{\prime }X$ as $X^{\prime }X=Q\Lambda Q^{\prime }$, we have that*

$$\left|\right|\beta_{MAP}\left(y\right)||=\left|\right|{\left(I+\left(\sigma^{2}/\tau^{2}\right)\Lambda^{-1}\right)}^{-1}Q^{\prime }b\left|\right|.$$

*Since $\left||b|\right|=\left|\right|Q^{\prime }b\left|\right|$ and $1/(1+\tau^{2}\lambda_{i}/\sigma^{2})<1$ for each i, this implies that $\beta_{MAP}\left(y\right)$ shrinks the MLE toward the prior mean of 0. When the columns of X are orthonormal, then $\beta_{MAP}\left(y\right)=r{\left(1+r\right)}^{-1}b$, where $r=\tau^{2}/\sigma^{2}$, and so the shrinkage is substantial unless $\tau^{2}$ is much larger than $\sigma^{2}.$ This shrinkage is often cited as a positive attribute of these estimates. Consider now the situation where the true value of β is some distance from the mean. In this case, it seems wrong to move β toward the prior mean; thus, it is not clear whether shrinking the MLE is necessarily a good thing, particularly as this requires giving up invariance.*

*Suppose that estimating the mean response $\psi =\Psi \left(\beta \right)=w^{\prime }\beta$ at w for the predictors is required. The prior distribution of ψ is $N\left(0,\sigma_{\psi }^{2}\right)=N\left(0,\tau^{2}w^{\prime }w\right)$ and the posterior distribution is $N\left(\psi_{MAP}\left(y\right),\sigma_{\psi ,post}^{2}\right)=N\left(w^{\prime }\beta_{MAP}\left(y\right),w^{\prime }\Sigma_{post}\left(\beta \right)w\right).$ Note the following relationships,*

$$\sigma_{\psi }^{2}-\sigma_{\psi ,post}^{2}=w^{\prime }\left(\tau^{2}I-\Sigma_{post}\right)w=\tau^{2}w^{\prime }Q^{\prime }\left(I-{\left(I+\left(\tau^{2}/\sigma^{2}\right)\Lambda \right)}^{-1}\right)Qw>0$$

*since $1/(1+\tau^{2}\lambda_{i}/\sigma^{2})<1$ for each i. Therefore, maximizing the ratio of the posterior to prior densities leads to*

(13)
$$\psi_{RB}\left(y\right)={\left(1-\sigma_{\psi ,post}^{2}/\sigma_{\psi }^{2}\right)}^{-1}\psi_{MAP}\left(y\right).$$

*Then $\sigma_{\psi }^{2}>\sigma_{\psi ,post}^{2}$ implies $|\psi_{RB}\left(y\right)|>|\psi_{MAP}\left(y\right)|.$ Note that when $\sigma_{\psi ,post}^{2}$ is much smaller than $\sigma_{\psi }^{2},$ in other words, the posterior is much more concentrated than the prior, then $\psi_{RB}\left(y\right)$ and $\psi_{MAP}\left(y\right)$ are very similar. In general, $\psi_{RB}\left(y\right)$ is not equal to $w^{\prime }b,$ the plug-in MLE of ψ, although it is the MLE from the integrated likelihood. Moreover, $\psi_{RB}\left(y\right)\to w^{\prime }b$ as $\tau^{2}\to \infty$, and when X has orthonormal columns, $\psi_{RB}\left(y\right)=w^{\prime }b.$*

*Suppose predicting a response z at the predictor value $w\in R^{k}$ is required. When $\beta ∼N_{k}\left(0,\tau^{2}I\right)$, the prior distribution of z is $z∼N\left(0,\sigma^{2}+\tau^{2}w^{\prime }w\right)=N\left(0,\sigma_{z}^{2}\right)$ and the posterior distribution is $N(\mu_{post}\left(z\right),\sigma_{post}^{2}\left(z\right))$, where we have that*

$$\mu_{post}\left(z\right)=w^{\prime }\beta_{post}\left(y\right),\;\sigma_{post}^{2}\left(z\right)=\sigma^{2}+w^{\prime }\Sigma_{post}w.$$

*To obtain $z_{RB}\left(y\right)$, it is necessary to maximize the ratio of the posterior to the prior densities of z; this leads to*

(14)
$$z_{RB}\left(y\right)={\left(1-\sigma_{post}^{2}\left(z\right)/\sigma_{prior}^{2}\left(z\right)\right)}^{-1}\mu_{post}\left(z\right).$$

*Note that $\sigma_{z}^{2}-\sigma_{post}^{2}\left(z\right)=\sigma_{z}^{2}\left(w^{\prime }\beta \right)-\sigma_{post}^{2}\left(w^{\prime }\beta \right)>0$; thus, $|z_{RB}\left(y\right)|>|\mu_{post}\left(z\right)|$, and $z_{RB}$ is further from the prior mean than $z_{MAP}\left(y\right)=\mu_{post}\left(z\right).$ Also, when $\sigma_{post}^{2}\left(z\right)$ is small, then $z_{RB}\left(y\right)$ and $z_{MAP}\left(y\right)$ are very similar. Finally, comparing (Equation 13) and (Equation 14), we have that*

$$z_{RB}\left(y\right)=\left(\sigma_{prior}^{2}\left(z\right)/\sigma_{post}^{2}\left(\psi \right)\right)w^{\prime }\psi_{RB}\left(y\right)=\left(1+\sigma^{2}/\tau^{2}\right)\psi_{RB}\left(y\right)$$

*and so $\psi_{RB}\left(y\right)$ at w is more dispersed than the $z_{RB}\left(y\right)$ estimate of the mean at w; this makes good sense as we have to take into account the additional variation due to prediction. By contrast, $w_{MAP}\left(y\right)=\psi_{MAP}\left(y\right).$*


## 5. Credible Regions and Hypothesis Assessment

Recall that a γ-relative-belief credible region for ψ=Ψ(θ) is given by $C_{\Psi ,\gamma }\left(x\right)=\left\{\psi :RB_{\Psi }\left(\psi \;|\;x\right)\geq c_{\gamma }\left(x\right)\right\}$, where $c_{\gamma }\left(x\right)=\sup\left\{c:\Pi_{\Psi }\left(RB_{\Psi }\left(\psi \;|\;x\right)\geq c\;|\;x\right)\geq \gamma \right\}.$ There is some arbitrariness in the choice of the greater than or equal sign to define the credible region as it could have been defined as $C_{\Psi ,\gamma }\left(x\right)=\left\{\psi :RB_{\Psi }\left(\psi \;|\;x\right)>c_{\gamma }\left(x\right)\right\}$, where $c_{\gamma }\left(x\right)=\inf\left\{c:\Pi_{\Psi }\left(RB_{\Psi }\left(\psi \;|\;x\right)\leq c\;|\;x\right)\leq 1-\gamma \right\}.$ In the latter case, $c_{\gamma }\left(x\right)$ is the (1−γ)-th quantile of the posterior distribution of the relative belief ratio. This definition has some advantages as using this implies that the plausible region satisfies $Pl_{\Psi }\left(x\right)=C_{\Psi ,\gamma }\left(x\right)$, where $\gamma =\Pi_{\Psi }\left(Pl_{\Psi }\left(x\right)\;|\;x\right).$ Also, the strength of the evidence concerning the hypothesis $H_{0}:\Psi \left(\theta \right)=\psi_{0}$ satisfies $Str_{\Psi }\left(\psi_{0}\;|\;x\right)=1-\Pi_{\Psi }\left(C_{\Psi ,\gamma }\left(x\right)\;|\;x\right)$ where $\gamma =1-Str_{\Psi }\left(\psi_{0}\;|\;x\right).$ The key point here is the close relationship between relative-belief credible regions, the plausible region, and the strength calculation. Thus, any decision-theoretic interpretation applicable to relative-belief credible regions also pertains to the plausible region and the strength of the evidence. Throughout this section, we will retain the definition for $C_{\Psi ,\gamma }\left(x\right)$ provided in Section 2.3.

Now consider the lowest posterior loss γ-credible regions that arise from the prior-based loss functions considered here.

**Theorem 5.** 
*Suppose that $\pi_{\Psi }\left(\psi \right)>0$ for every ψ∈Ψ(Θ), where Ψ(Θ) is finite with $\nu_{\Psi }$ equal to the counting measure. Then $C_{\Psi ,\gamma }\left(x\right)$ is a γ-lowest posterior loss-credible region for the loss function $L_{RB}.$*


**Proof.** From (Equation 2) and (Equation 6), the γ-lowest posterior loss-credible region is given by
$$D_{\gamma }\left(x\right)=\left\{\psi :RB_{\Psi }\left(\psi \;|\;x\right)\geq \int_{\Psi }RB_{\Psi }\left(\zeta \;|\;x\right)\;\nu_{\Psi }\left(d\zeta \right)-d_{\gamma }\left(x\right)\right\}$$
and $d_{\gamma }\left(x\right)=\sup\{d:\Pi_{\Psi }\left(\left\{\psi :r\left(\psi \;|\;x\right)\leq d\;|\;x\right)\geq \gamma \right\}.$ As $\int_{\Psi }RB_{\Psi }\left(z\;|\;x\right)\;\nu_{\Psi }\left(dz\right)$ is independent of ψ it is clearly equivalent to define this region via $C_{\Psi ,\gamma }\left(x\right)=\left\{\psi :RB_{\Psi }\left(\psi \;|\;x\right)\geq c_{\gamma }\left(x\right)\right\},$ namely, $D_{\gamma }\left(x\right)=C_{\Psi ,\gamma }\left(x\right).$ □

Now consider the case where Ψ is countable and we use the loss function $L_{RB,\eta }.$ Following the proof of Theorem 5, we see that a γ-lowest posterior loss region takes the form,
$$D_{\eta ,\gamma }\left(x\right)=\left\{\psi :\pi_{\Psi }\left(\psi \;|\;x\right)/\max\left(\eta ,\pi_{\Psi }\left(\psi \right)\right)\geq d_{\eta ,\gamma }\left(x\right)\right\},$$
where $d_{\eta ,\gamma }\left(x\right)=\sup\left\{d:\Pi_{\Psi }\left(\left\{\psi :\pi_{\Psi }\left(\psi \;|\;x\right)/\max\left(\eta ,\pi_{\Psi }\left(\psi \right)\right)\geq d\;|\;x\right)\right\}\geq \gamma \right\}.$

**Theorem 6.** 
*Suppose that $\pi_{\Psi }\left(\psi \right)>0$ for every ψ∈Ψ, the Ψ is countable with $\nu_{\Psi }$ equal to counting measure. For the loss function $L_{RB,\eta }$, $C_{\Psi ,\gamma }\left(x\right)\subset \lim\inf_{\eta \to 0}D_{\eta ,\gamma }\left(x\right)$ whenever γ is such that $\Pi_{\Psi }\left(C_{\Psi ,\gamma }\left(x\right)\;|\;x\right)=\gamma$, and $\lim\sup_{\eta \to 0}D_{\eta ,\gamma }\left(x\right)\subset C_{\Psi ,\gamma^{\prime }}\left(x\right)$ whenever $\gamma^{\prime }>\gamma$ and $\Pi_{\Psi }\left(C_{\Psi ,\gamma^{\prime }}\left(x\right)\;|\;x\right)=\gamma^{\prime }.$*


While Theorem 6 does not establish the exact convergence $\lim_{\eta \to 0}D_{\eta ,\gamma }\left(x\right)=C_{\gamma }\left(x\right)$, it is likely that this does hold under quite general circumstances due to the discreteness. Theorem 6 shows that limit points of the class of sets $D_{\eta ,\gamma }\left(x\right)$ always contain $C_{\Psi ,\gamma }\left(x\right)$ and their posterior probability content differs from γ by at most $\gamma^{\prime }-\gamma$, where $\gamma^{\prime }>\gamma$ is the next largest value for which we have exact content.

Now, consider the continuous case with a regular discretization. For $S^{*}\subset \Psi_{\lambda }=\left\{\psi_{\lambda }\left(\psi \right):\psi_{\lambda }\left(\psi \right)\in B_{\lambda }\left(\psi \right)\right\},$ namely, $S^{*}$ is a subset of a discretized version of Ψ(Θ), we define the *un-discretized* version of $S^{*}$ to be $S=\cup_{\psi \in S^{*}}B_{\lambda }\left(\psi \right).$ Now, let $C_{\Psi ,\lambda ,\gamma }^{*}\left(x\right)$ be the γ-relative belief region for the discretized problem and let $C_{\Psi ,\lambda ,\gamma }\left(x\right)$ be its un-discretized version. Note that in a continuous context, we will consider two sets as equal if they differ only by a set of measure 0 with respect to $\Pi_{\Psi }.$ The following result says that a γ-relative belief-credible region for the discretized problem, after un-discretizing, converges to the γ-relative belief region for the original problem.

**Theorem 7.** 
*Suppose that $\pi_{\Psi }$ is positive and continuous, there is regular discretization of Ψ(Θ) and $RB_{\Psi }\left(\psi \;|\;x\right)$ has a continuous posterior distribution. Then, $\lim_{\lambda \to 0}C_{\Psi ,\lambda ,\gamma }\left(x\right)=C_{\Psi ,\gamma }\left(x\right).$*


While Theorem 7 is interesting in its own right, it can also be used to prove that relative belief regions are limits of the lowest posterior loss regions.

Let $D_{\eta ,\lambda ,\gamma }^{*}\left(x\right)$ be the γ-lowest posterior loss region obtained for the discretized problem using loss function (Equation 12), and let $D_{\eta ,\lambda ,\gamma }\left(x\right)$ be the un-discretized version.

**Theorem 8.** 
*Suppose that $\pi_{\Psi }$ is positive and continuous, we have a regular discretization of Ψ, and $RB_{\Psi }\left(\psi \;|\;x\right)$ has a continuous posterior distribution. Then, we have that*

$$C_{\Psi ,\gamma }\left(x\right)=\lim_{\lambda \to 0}\underset{\eta \to 0}{\lim\inf}D_{\Psi ,\gamma }\left(x\right)=\lim_{\lambda \to 0}\underset{\eta \to 0}{\lim\sup}D_{\Psi ,\gamma }\left(x\right).$$



In [18,19], additional properties of relative belief regions are developed. For example, it has been shown that a γ-relative belief region $C_{\Psi ,\gamma }\left(x\right)$ for ψ, satisfying $\Pi_{\Psi }\left(C_{\Psi ,\gamma }\left(x\right)\;|\;x\right)=\gamma$, minimizes $\Pi_{\Psi }\left(B\right)$ among all (measurable) subsets of Ψ satisfying $\Pi_{\Psi }\left(B\;|\;x\right)\geq \gamma .$ So, a γ-relative belief region is the smallest among all γ-credible regions for ψ, where the size is measured using the prior measure. This property has several consequences. For example, the prior probability that a region B(x)⊂Ψ(Θ) contains a false value from the prior is given by $\int_{\Theta }\int_{\Psi }F_{\theta }\left(\psi \in B\left(x\right)\right)\;\Pi_{\Psi }\left(d\psi \right)\;\Pi \left(d\theta \right)$, where a false value is a value of $\psi ∼\Pi_{\Psi }$ generated independently of $\left(\theta ,x\right)∼\Pi_{\Psi }\times F_{\theta }.$ It can be demonstrated that a γ-relative belief region minimizes this probability among all γ-credible regions for ψ and is always unbiased in the sense that the probability of covering a false value is bounded above by γ. Furthermore, a γ-relative belief region maximizes the relative belief ratio $\Pi_{\Psi }\left(B\;|\;x\right)/\Pi_{\Psi }\left(B\right)$ and the Bayes factor $\Pi_{\Psi }\left(B\;|\;x\right)\Pi_{\Psi }\left(B^{c}\right)/\Pi_{\Psi }\left(B^{c}\;|\;x\right)\Pi_{\Psi }\left(B\right)$ among all regions B⊂Ψ with $\Pi_{\Psi }\left(B\right)=\Pi_{\Psi }\left(C_{\Psi ,\gamma }\left(x\right)\;|\;x\right).$

While the results in this section focus on obtaining credible regions for parameters, similar results can be proven for the construction of prediction regions.

## 6. Conclusions

Relative belief inferences are based on a clear characterization of statistical evidence and are closely related to likelihood inferences. This, along with their invariance and optimality properties, positions these as prime candidates for appropriate inferences in Bayesian contexts. This paper shows that relative belief inferences also arise naturally in a decision-theoretic formulation using loss functions based on the prior. So, relative belief inferences represent a degree of unification between the evidential and decision-theoretic approaches to deriving statistical inferences.

## Figures and Tables

**Table 1 entropy-26-00786-t001:** Conditional prior probabilities of misclassification for $c_{MAP}$ and $c_{RB}$ for various values of β in Example 2 when α=1, μ=1, and *n* = 10.

β	$M_{0}\left(c_{\mathrm{MAP}}\neq 0\right)+M_{1}\left(c_{\mathrm{MAP}}\neq 1\right)$	$M_{0}\left(c_{\mathrm{RB}}\neq 0\right)+M_{1}\left(c_{\mathrm{RB}}\neq 1\right)$
1	0.386+0.390=0.776	0.386+0.390=0.776
14	0.002+0.975=0.977	0.285+0.380=0.665
32	0.000+0.997=0.997	0.292+0.349=0.641
100	0.000+1.000=1.000	0.300+0.324=0.624

## Data Availability

Data is contained within the article.

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
