# Peer review of "Relative Belief Inferences from Decision Theory"

_entropy, 2024, doi:10.3390/e26090786_

Round 1

Reviewer 1 Report

Comments and Suggestions for Authors

Comments on the Quality of English Language

Author Response

See attached,

Reviewer 2 Report

Comments and Suggestions for Authors

see pdf

Round 2

Reviewer 1 Report

Comments and Suggestions for Authors

The revision has answered most of my concern. One minor issue that was probably misunderstood is related to equation (1). For a fixed data and model, MAP estimator can be made to be any value by simply changing the dominating measure mu to 

h dmu which results in the posterior density being p/h.

Then all I need to do is to make sure that h is sufficiently close to 0 near the point I want to have the MAP. 

The form of equation (1) implicitly presumes a certain kind of dominating measure, that would made the volume of the balls of the same radius the same regardless of location.. This needs to be made clear.

Comments on the Quality of English Language

None